# Drug-Associated Parosmia: New Perspectives from the WHO Safety Database

**DOI:** 10.3390/jcm11164641

**Published:** 2022-08-09

**Authors:** Diane Merino, Alexandre Olivier Gérard, Susanne Thümmler, Nouha Ben Othman, Delphine Viard, Fanny Rocher, Alexandre Destere, Elise Katheryne Van Obberghen, Milou-Daniel Drici

**Affiliations:** 1Department of Psychiatry, University Hospital of Nice, 06000 Nice, France; 2Department of Pharmacology and Pharmacovigilance Centre of Nice, University Hospital of Nice, 06000 Nice, France; 3Department of Nephrology-Dialysis-Transplantation, University Hospital of Nice, 06000 Nice, France; 4CoBTek, FRIS, Université Côte d’Azur, 06100 Nice, France; 5Children’s Hospitals of Nice CHU-Lenval, 06200 Nice, France; 6Department of Neurology, University Hospital of Nice, 06000 Nice, France

**Keywords:** smell, olfaction, parosmia, adverse drug reaction, clinical epidemiology, pharmacology

## Abstract

Parosmia is a qualitative distortion of smell perception. Resulting from central causes, sinonasal diseases, and infections, parosmia has also been associated with medications. Therefore, we aimed to investigate potential signals for drugs associated with parosmia. VigiBase^®^ (the WHO pharmacovigilance database) was queried for all reports of “Parosmia” (MedDRA Preferred Term), registered up to 23 January 2022. Disproportionality analysis relied on the reporting odds ratio and the information component. A signal is detected when the lower end of the 95% confidence interval of the information component is positive. We found 14,032 reports of parosmia, with a median patient age of 53 years. Most reported drugs were antiinfectives, among which COVID-19 vaccines accounted for 27.1% of reports. Antibiotics and corticosteroids were involved in 6.8% and 4.6% of reports. Significant disproportionate reporting was detected for corticosteroids, antibiotics, drugs used in nicotine dependence, COVID-19 and HPV vaccines, serotonin–norepinephrine reuptake inhibitors (SNRI), and incretin mimetics. We suggest potential safety signals involving nicotine replacement therapies and vaccines. We also highlight the potential role of less suspected classes, such as SNRIs and incretin mimetics. An iatrogenic etiology should be evoked when parosmia occurs, especially in the elderly.

## 1. Introduction

Parosmia (or troposmia) is a qualitative olfactory dysfunction, characterized by a distortion of smell perception in the presence of a known source [1,2]. Most often unpleasant, parosmia is often associated with an alteration of taste and quality of life. With an estimated prevalence of around 4%, parosmia mostly affects females between 20 and 29 years old [3,4]. In the absence of standardized criteria, the diagnosis of parosmia can rely on the Sniffin’ Sticks test [5].

Physiologically, odor molecules are first detected in the nose, by the olfactory epithelium. The interaction of odorants with their receptors initiates an electrical signal. The signal reaches the olfactory bulb and then the cerebral regions that will lead to awareness of the odor perception and to behavioral, emotional, and autonomic responses to odor [6,7]. 

Parosmia may result from dysfunction or loss of olfactory neurons. These peripheral causes include post-upper respiratory tract infections (URIs) and sinonasal diseases. Besides, the central theory implies a defect in the cerebral integrative or interpretive centers, which would lead to a distorted and/or unpleasant smell perception (decreased bulb volume [8,9], head trauma) [1,8]. Furthermore, several studies suggest an association between parosmia and different drugs, such as antibiotics and vaccines [10,11,12].

To characterize the potential class–effect associations and possible mechanisms involved, we aimed to investigate potential safety signals for drugs associated with parosmia. We, therefore, conducted a disproportionality analysis, relying on the international pharmacovigilance database of the World Health Organization (WHO).

## 2. Materials and Methods

### 2.1. Data Source

Since 1978, the Uppsala Monitoring Centre (UMC) is mandated by the WHO to oversee drug safety. The UMC is an independent center for drug safety and scientific research, which aims to collect evidence about harm to patients and to identify safety signals [13]. Data issued by the 172 national pharmacovigilance network members and pharmaceutical companies are collected by the UMC database. VigiBase^®^ [14] (the WHO pharmacovigilance database) gathers spontaneous reports and ensures the preservation of the anonymity of both patients and notifiers.

Sociodemographic characteristics (age, sex, area of residence, notifier’s country) and details concerning the reported effect (suspected drugs, concomitant drugs, adverse drug reaction, date of occurrence, and seriousness) are collected in the database. In pharmacovigilance, an adverse drug reaction (ADR) is deemed serious if it justified an hospitalization or the prolongation of an hospitalization, caused a congenital malformation, resulted in persistent or significant disability or incapacity, was life-threatening, resulted in death, or required significant medical intervention to prevent one of these outcomes [15,16].

### 2.2. Query

VigiBase^®^ was queried for all reports of the preferred term (PT) “Parosmia” (Medical Dictionary for Regulatory Activities, MedDRA 24.1 [17]), registered between 14 November 1967 (first reports in VigiBase^®^) and 23 January 2022. A PT expresses a single medical concept in the most clinically accurate way. Then, all active ingredients deemed suspect or interacting in 30 or more cases were ranked by absolute number of cases. Following the same method, we also queried VigiBase^®^ for all reports corresponding to the PT “Hyposmia”.

Quantitative variables were described in terms of medians with interquartile range (IQR) or means with standard deviations (±SD). Qualitative variables were described with proportions. Characteristics of patients were compared using Pearson’s chi-square test for categorical variables, or Student’s *t*-test for normally distributed continuous variables. *p* < 0.05 was considered to be statistically significant. Statistical analyses were performed using GraphPad Prism version 8.0.2.

### 2.3. Disproportionality Analysis

Subsequently, a disproportionality analysis was conducted to mitigate the impact of potential overreporting of cases involving COVID-19 vaccines. Hence, we sought a potential pharmacovigilance signal for all drugs involved in parosmia reports. We applied the same method to hyposmia reports. This disproportionality analysis was based on the reporting odds ratio (ROR) and the information component (IC).

The ROR, as an approximate of the odds ratio (used in case–control studies), is estimated in case–non-case studies to assess the strength of disproportionality. An ROR equal to 1 states the absence of signal: the ADR is similarly reported with the drug of interest as with the other drugs. Conversely, an ROR greater than 1 suggests a signal, as cases are more frequently reported with the drug of interest than with the others. The higher the ROR, the stronger the association. The precision of the approximate ROR is reflected by a 95% confidence interval (95% CI). Consequently, an ROR is considered to be statistically significant when the lower bound of its 95% CI is greater than 1 [18].

The IC compares observed and expected values for the combination of a given drug and an ADR to yield associations between them. It helps in lowering the risk of false-positive signals, especially if the chosen ADR has a very low expected frequency in the database (hence mechanically increasing the ROR). The positivity of the IC reveals the superiority of the number of observed reports over the number of expected reports. The IC025 is the bottom end of the 95% CI of the information component. For UMC, a positive IC025 is required to statistically confirm the detection of a signal [14].

In this disproportionality analysis, relevant associations were chosen using IC025. Then, active ingredients showing a significant association with parosmia were ranked according to their absolute number of reports. Lastly, the strength of each drug–effect association was investigated using the respective RORs.

## 3. Results

### 3.1. Characteristics of the Reports

As of 23 January 2022, among 29,978,090 reports registered in VigiBase^®^, 14,032 were cases of parosmia (PT). More than half of the reports (7475, 53.3%) originated from the United States. When mentioned, the notifier was mostly a consumer (5324 reports, 37.9%). Parosmia was considered to be serious in 3216 (22.9%) notifications. Most patients were females (8972, 63.9%). The most represented age range was the 45-to-64-year group, with 4740 (33.8%) reports, and the median age was 53 years (IQR: 40–63). The median time to onset was 11 days (IQR: 1–83). Most patients with available outcomes did not recover (3596, 53.3%), while 2256 (33.4%) patients recovered, 690 (10.2%) were recovering, and 206 (3.1%) recovered with sequelae. “Dysgeusia” (3157, 22.5%), “nausea” (2252, 16.0%), and “headache” (1981, 14.1%) were the most frequent co-reported MedDRA clinical terms. Detailed characteristics of the reports are provided in Table 1.

### 3.2. Active Ingredients Ranked by Absolute Number of Reports

The launch of COVID-19 vaccines was associated with a surge in reports of parosmias, without modifying the underlying trend of reporting for other drugs, as compared to previous years (Figure 1). As shown in Table 2, patients having received a COVID-19 vaccine were significantly older (mean age: 53.2 ± 16.3 vs. 47.1 ± 15.5; *p* < 0.001) and were more likely to be females (71.5% vs. 65.3%; *p* < 0.001) than patients treated with other drugs. Furthermore, cases involving COVID-19 vaccines were less frequently reported by healthcare professionals as compared to cases involving other drugs (18.0% vs. 50.8%; *p* < 0.001). Among parosmia cases involving COVID-19 vaccines, most patients with available outcomes did not recover (1124, 51.1%), while 757 (34.4%) patients recovered, 284 (25.2%) were recovering, and 32 (2.8%) recovered with sequelae.

Accordingly, “Antiinfectives for systemic use” (J) was the most represented drug class in the Anatomical Therapeutic Chemical (ATC) classification system, with 5927 (42.2%) notifications. Figure 2 summarizes the main ATC classes involved in parosmia reports, and a more comprehensive list of active ingredients is provided in Appendix A. COVID-19 vaccines were the active ingredients most frequently reported as suspect or interacting, with more than one-quarter (3796, 27.1%) of the reports (tozinameran, elasomeran, AZD1222, JNJ-78436735). The human papillomavirus (HPV) vaccine was also reported in 106 (0.8%) cases. 

Next, antibiotics accounted for 948 (6.8%) reports, mostly involving macrolides (565, 4.1%) and quinolone antibiotics (305; 2.1%). Corticosteroids were involved in 630 (4.6%) cases (fluticasone, beclomethasone, mometasone, flunisolide, budesonide, triamcinolone). When available, the route of administration for corticosteroids was nasal or inhaled in 91.7% of reports. 

Drugs used in nicotine dependence accounted for 378 (2.7%) cases (varenicline, bupropion, nicotine). Antidepressants were represented by the serotonin–norepinephrine reuptake inhibitors (SNRIs) duloxetine (82, 0.6%) and venlafaxine (49, 0.3%) but also by the selective serotonin reuptake inhibitor (SSRI) paroxetine (66, 0.5%). Incretin mimetics were also involved (exenatide, dulaglutide, liraglutide).

### 3.3. Disproportionality Analysis

Significant disproportionality was found for 47 active ingredients with an IC025 > 0, ranked according to their absolute number of cases, as shown in Table 3.

Flunisolide reached the strongest ROR (144.5; 95% CI 115.5–180.5). Other corticosteroids, namely, beclomethasone (25.8; 95% CI 21.9–30.4), mometasone (14.2; 95% CI 11.7–17.2), fluticasone (11.9; 95% CI 10.5–13.6), triamcinolone (3.9; 95% CI 2.8–5.3), and budesonide (3.7; 95% CI 2.8–4.9), were also characterized by significant RORs.

Among antibiotics, macrolides such as roxithromycin (19.8; 95% CI 16.6–23.6), clarithromycin (11.7; 95% CI 10.4–13.2), and azithromycin (4.3; 95% CI 3.6–5.0) had significant disproportionate reporting. RORs of quinolone antibiotics, represented by ofloxacin (7.4; 95% CI 6.0–9.1), moxifloxacin (4.1; 95% CI 3.4–5.1), and ciprofloxacin (2.0; 95% CI 1.7–2.4) were also significant. Doxycycline was characterized by an ROR of 4.3 (95% CI 3.4–5.3).

Drugs used in nicotine replacement therapy (NRT), including varenicline, bupropion, and nicotine, showed RORs of 5.8 (95% CI 5.1–6.6), 2.4 (95% CI 1.9–3.0), and 1.5 (95% CI 1.2–2.0), respectively.

COVID-19 vaccines were characterized by a significant ROR (3.3; 95% CI 3.2–3.4). Tozinameran reached the strongest ROR (3.3; 95% CI 3.2–3.5), followed by JNJ-78436735 (3.2; 95% CI 2.8–3.7), elasomeran (2.8; 95% CI 2.6–3.1), and AZD1222 (2.3; 95% CI 2.1–2.4). Furthermore, the HPV vaccine also had a significant ROR (2.0; 95% CI 1.6–2.4). However, reports of parosmia following influenza vaccination (Appendix A) were not subject to any disproportionality signal.

Among antidepressants, duloxetine reached the strongest ROR (2.5; 95% CI 2.0–3.1), followed by paroxetine (1.9; 95% CI 1.5–2.4) and venlafaxine (1.5; 95% CI 1.1–1.9).

Then, the incretin mimetics exenatide (2.0; 95% CI 1.5–2.5), dulaglutide (1.7; 95% CI 1.3–2.4), and liraglutide (1.6; 95% CI 1.1–2.3) also showed disproportionate reporting.

Drugs showing disproportionate reporting with an absolute number of cases greater than 30 are presented in Table 3.

### 3.4. Hyposmia Reports

As of January 23, 2022, 1741 cases of hyposmia (PT) were gathered in VigiBase^®^. The most represented ATC drug classes were antiinfectives (525; 30.2%) and respiratory system medications (512; 29.4%). In fact, the three leading drugs in terms of absolute number of cases were COVID-19 vaccines (384; 22.4%), oxymetazoline (197, 11.3%), and fluticasone (56; 3.2%). These drugs were subject to a disproportionality drug signal. Among cases involving COVID-19 vaccines that displayed available outcomes, 179 patients recovered or were recovering (89.4%), 89 patients did not recover (39.2%), and 5 patients (2.2%) recovered with sequelae.

## 4. Discussion

This analysis of the WHO pharmacovigilance database suggests an association between several drug classes and parosmia, with a middle-aged-female predominance, in contrast to previous studies [3]. While our analysis of VigiBase^®^ concurred with gender distribution, patients with drug-induced parosmia were older than expected. The fact that antidepressants [19], nicotine replacements [20], and antidiabetic drugs are more frequently prescribed in older populations [21] may explain this discrepancy.

Over one-fifth of parosmia cases retrieved were considered to be serious in our analysis. Patients with parosmia have a diminished quality of life, present more daily-life complaints, and suffer frequently from taste alteration [12]. Such physical and psychological impact may lead to depression and weight loss [22].

While the underlying mechanism remains poorly understood, macrolides and quinolone antibiotics stood out in terms of disproportionality and number of cases. Indeed, a class–effect relationship has been suggested previously [10], and the labels of these drugs mention smell disturbances [23,24]. Six corticosteroids stood out in terms of disproportionality, representing a strong association of this class with parosmia. Their labels indicate loss of smell among possible ADRs [25,26]. Nevertheless, oral and nasal corticosteroids are a controversial treatment in some cases of anosmia, especially when they are COVID induced [27,28,29,30]. The presence of glucocorticoid receptors in the olfactory mucosa suggests a functional role for corticosteroids in smell regulation, which might be impaired in parosmia [31,32]. Accordingly, almost all cases of corticosteroid-associated parosmia reported in VigiBase^®^ involved inhaled delivery. Indeed, URIs and sinonasal diseases, which are common causes of parosmia [12], are sometimes treated by antibiotics and corticosteroids. However, the absence of signal for some first-line treatments of these affections, such as amoxicillin, strengthens the hypothesis of class–effect associations for macrolides and quinolone antibiotics.

Interestingly, COVID-19 vaccines accounted for more than one-quarter of the parosmia reports of the WHO drug safety database, while there is growing evidence concerning COVID-19-related parosmia [33,34,35]. Indeed, the number of parosmia reports dramatically increased in 2021 (Figure 1), due to reports involving COVID-19 vaccines. While parosmia frequently appears following anosmia or hyposmia [35], we were not able to detect any potential smell loss preceding the occurrence of parosmia among cases included in VigiBase^®^. However, these reports were more frequently reported by non-healthcare professionals (when compared with other drugs), which may alter their overall reliability. This transient symptom has been attributed to a vaccine-induced inflammatory reaction localized in the olfactory neuroepithelium in a few reports [36,37,38,39]. Furthermore, our study reinforces previous findings about the HPV vaccine [40], suggesting a possible class–effect association. By contrast with former studies, we did not suggest any signal involving influenza vaccines [41,42].

Diabetes [43] is usually associated with neurosensory disorders, thus leading to smell impairment. Incretin mimetics stood out in the analysis of parosmia reports. Exenatide, dulaglutide, and liraglutide are glucagon-like peptide 1 (GLP-1) receptor agonists, which potentiate glucose-stimulated insulin secretion by the pancreas [44]. A previous study revealed a high distribution of insulin receptors in the olfactory bulb of rats, so that insulin variations may alter olfactory processing [45]. Likewise, the role of insulin in triggering anosmia has formerly been suggested [11]. In addition, the precursor and the receptor mRNAs of GLP-1 have been found among olfactory bulb cells, so that GLP-1 may help in mediating sensory information from the olfactory epithelium to the brain [46,47,48]. 

Parosmia has been correlated to the severity of depression [49]. Our findings about SNRIs may concur with the central theory of parosmia, involving the dysfunction of cerebral integrative centers [1]. Parosmic patients have been found to exhibit stronger activation in the thalamus and the putamen, which are related to attention and disgust, respectively [8]. Similarly, antidepressants such as SNRIs modulate their activation, depending on the polarity of the stimuli [50] and, therefore, might interact with odor interpretation. By inhibiting norepinephrine reuptake, SNRIs may impact the activation of the amygdala, which receives information from olfactory pathways [7,51]. Paroxetine inhibits norepinephrine reuptake to a lesser extent, but this feature sets it apart from other SSRIs [52]. These findings may support a noradrenergic mechanism for antidepressant-induced parosmia.

Tobacco smoking is a well-known cause of olfactory dysfunction [53]. More surprisingly, several drugs used in NRT showed a significant disproportionate reporting with parosmia, which suggests the existence of a common mechanism. Varenicline, an agonist for the α4β2 nicotinic acetylcholine receptor subtype (nACh), enhances dopamine release in the nucleus accumbens [54]. The latter being closely related to the olfactory tubercle, deregulation in olfaction could be expected [55]. The mechanism of parosmia induced by nicotine and bupropion, a norepinephrine–dopamine reuptake inhibitor (NDRI) and antagonist of several nACh receptors, may be similar [56,57]. 

This global analysis of drug-associated parosmia reports from the WHO safety database highlights the different classes involved. However, its limitations, such as reporting bias, incomplete data, and lack of follow-up are inherent to post-marketing pharmacovigilance studies. As parosmia lacks a specific diagnostic tool, and as differential diagnosis might be missing in some cases, coding heterogeneity may be a prevalent limitation. Further, only 14,032 cases of parosmia were recorded in VigiBase^®^ since 1967 (first reports in the database), suggesting a very significant under-reporting, usual in pharmacovigilance. Indeed, irrespective of the ADR, a meta-analysis [58] retrieved a median under-reporting rate of 94% in spontaneous reporting systems. It seems all the more important to heighten awareness of drug-induced parosmia. Some patients may have trouble distinguishing a consequence of their underlying disease from a potential ADR (e.g., corticosteroids in smell loss). Likewise, the notion of COVID-19 infection (or not) before vaccination of patients could have helped to assess a potential confounding between parosmia and COVID-19 vaccine, but this information was not available in VigiBase^®^. 

In any case, pharmacovigilance studies are exploratory and aim to raise awareness by suggesting potential safety signals. The causal link between parosmia and serotonin–norepinephrine reuptake inhibitors, COVID-19 and HPV vaccines, drugs used in nicotine replacement therapy, and incretin mimetics must be further confirmed.

## 5. Conclusions

In this study, based on a comprehensive analysis of parosmia reports from the WHO safety database, we suggest potential pharmacovigilance signals involving several drug classes, mainly comprising nicotine replacement therapies and vaccines. We also highlight the potential role of less suspected classes, such as serotonin–norepinephrine reuptake inhibitors and incretin mimetics. Polypharmacy notoriously concerns older individuals, therefore increasing their risk of drug-related parosmia [59]. An iatrogenic etiology should be especially evoked when parosmia occurs in the elderly. Parosmia being a major cause of quality-of-life impairment, identification of drug triggers may help the clinician in the perspective of treatment adjustments.

## Figures and Tables

**Figure 1 jcm-11-04641-f001:**
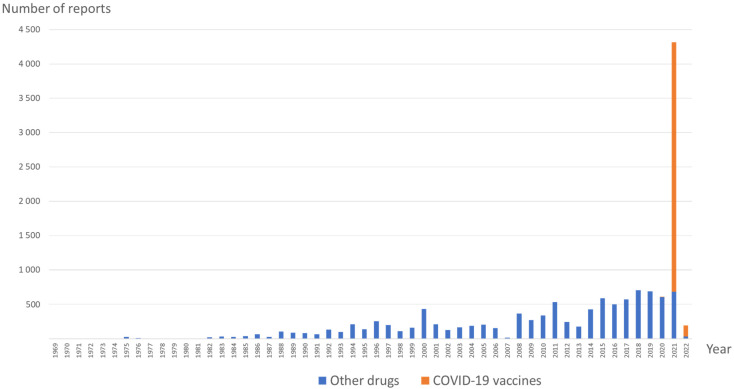
Prevalence of parosmia reports involving COVID-19 vaccines and other drugs over time.

**Figure 2 jcm-11-04641-f002:**
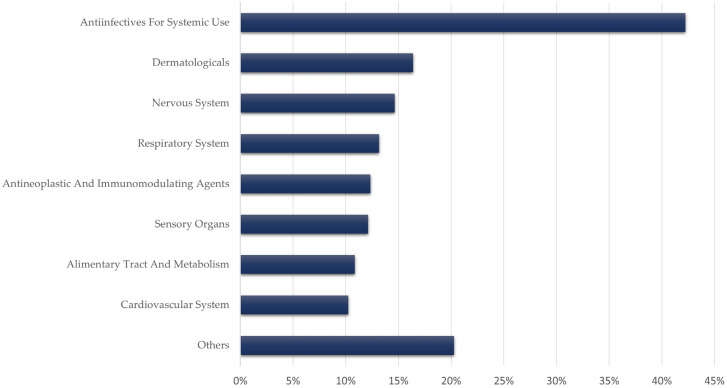
Main Anatomical Therapeutic Chemical (ATC) classes involved in parosmia reports.

**Table 1 jcm-11-04641-t001:** Characteristics of the reports of patients with parosmia.

Characteristics	Number (%)
**Total**	**14,032 (100)**
**Sex**	
Female	8972 (63.9)
Male	4441 (31.6)
Unknown	619 (4.4)
**Country**	
United States of America	7475 (53.3)
Germany	1354 (9.6)
United Kingdom	1202 (8.6)
Netherlands	639 (4.6)
France	351 (2.5)
Australia	324 (2.3)
Canada	301 (2.1)
Korea	288 (2.1)
Sweden	247 (1.8)
Italy	221 (1.6)
Spain	186 (1.3)
Denmark	147 (1.0)
Norway	124 (0.9)
Switzerland	105 (0.7)
Japan	96 (0.7)
Belgium	91 (0.6)
Finland	87 (0.6)
Austria	84 (0.6)
Poland	57 (0.4)
Ireland	53 (0.4)
China	51 (0.4)
Brazil	43 (0.3)
Czechia	42 (0.3)
Croatia	38 (0.3)
Romania	31 (0.2)
New Zealand	30 (0.2)
Others	365 (2.6)
**Reporter Qualification**	
**Healthcare Professional**	**4231 (30.1)**
Physician	2636 (18.8)
Pharmacist	784 (5.6)
Other Health Professional	811 (5.8)
**Others**	**5349 (38.1)**
Lawyer	25 (0.2)
Consumer	5324 (37.9)
**Unknown**	**4452 (31.7)**

**Table 2 jcm-11-04641-t002:** Characteristics of the reports of patients with parosmia (COVID-19 vaccines vs. other drugs).

	Other Drugs’ Reports	COVID-19 Vaccines’ Reports	*p* Value
Reported by a healthcare professional (%)	50.8	18.0	<0.001
Female (%)	65	72	<0.001
Age (Mean ± SD)	47.1 ± 15.5	53.2 ± 16.3	<0.001

SD: standard deviation.

**Table 3 jcm-11-04641-t003:** Disproportionality analysis for reports of parosmia.

Active Ingredient	ROR	95% CI	Number of Cases (%)
Flunisolide	144.5	115.6–180.5	83 (0.6)
Oxymetazoline	41.4	33.9–50.5	99 (0.7)
Beclometasone	25.8	21.9–30.4	145 (1.1)
Ipratropium	25.0	20.7–30.1	113 (0.8)
Roxithromycin	19.8	16.6–23.6	125 (0.9)
Mometasone	14.2	11.7–17.2	106 (0.8)
Cromoglicic acid	12.2	8.7–17.0	34 (0.2)
Fluticasone	11.9	10.5–13.6	223 (1.6)
Clarithromycin	11.7	10.4–13.2	288 (2.1)
Terbinafine	9.3	7.8–11.2	121 (0.9)
Ofloxacin	7.4	6.0–9.1	91 (0.6)
Varenicline	5.8	5.1–6.6	245 (1.7)
Peginterferon alfa-2b	4.8	3.7–6.3	55 (0.4)
Lovastatin	4.6	3.4–6.4	39 (0.3)
Azithromycin	4.3	3.6–5.0	152 (1.1)
Doxycycline	4.3	3.4–5.3	78 (0.6)
Moxifloxacin	4.1	3.4–5.1	95 (0.7)
Triamcinolone	3.9	2.8–5.3	38 (0.3)
Budesonide	3.7	2.8–4.9	45 (0.3)
Telaprevir	3.4	2.6–4.4	52 (0.4)
**COVID-19 Vaccine**	**3.3**	**3.2–3.4**	**3796 (27.1)**
Tozinameran	3.3	3.2–3.5	2103 (55.4)
JNJ 78436735	3.2	2.8–3.7	191 (5.0)
Elasomeran	2.8	2.6–3.1	745 (19.6)
AZD1222	2.3	2.1–2.4	724 (19.1)
Captopril	3.1	2.3–4.1	43 (0.3)
Pirfenidone	2.8	2.0–3.7	41 (0.3)
Ribavirin	2.5	2.1–3.0	121 (0.9)
Duloxetine	2.5	2.0–3.1	82 (0.6)
Bupropion	2.4	1.9–3.0	81 (0.6)
Metamizole	2.4	1.7–3.4	32 (0.2)
Peginterferon alfa-2a	2.2	1.7–2.9	54 (0.4)
Budesonide; Formoterol	2.2	1.6–3.0	35 (0.2)
Clonazepam	2.1	1.5–2.8	39 (0.3)
Ciprofloxacin	2.0	1.7–2.4	119 (0.8)
Salbutamol	2.0	1.6–2.5	78 (0.6)
Exenatide	2.0	1.5–2.5	71 (0.5)
HPV vaccine	2.0	1.6–2.4	106 (0.8)
Paroxetine	1.9	1.5–2.4	66 (0.5)
Cetirizine	1.8	1.2–2.5	31 (0.2)
Dulaglutide	1.7	1.3–2.4	40 (0.3)
Metronidazole	1.7	1.3–2.2	60 (0.4)
Palbociclib	1.6	1.2–2.2	47 (0.3)
Liraglutide	1.6	1.1–2.3	30 (0.2)
Citalopram	1.6	1.1–2.2	31 (0.2)
Nicotine	1.5	1.2–2.0	52 (0.4)
Venlafaxine	1.5	1.1–1.9	49 (0.3)
Teriparatide	1.3	1.1–1.6	95 (0.7)

ROR: reporting odds ratio; CI: confidence interval.

## Data Availability

The data that support the findings of this study are available from Uppsala Monitoring Centre (UMC), but restrictions apply to the availability of these data, which were used under license for the current study, and so are not publicly available. Access to VigiBase^®^ is available without fees to F.R. Data are, however, available from the authors upon reasonable request and with permission of UMC.

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
