# Peer review of "Drug-Associated Parosmia: New Perspectives from the WHO Safety Database"

_jcm, 2022, doi:10.3390/jcm11164641_

Round 1
Reviewer 1 Report
Parosmia is a common problem in the era of the COVID-19 pandemic with poor results in the short-term treatment and follow-up. To investigate potential signals for drugs associated with parosmia, authors used VigiBase, the richest source of pharmacovigilance data in the world and perform disproportionality analysis. Significant disproportionate reporting was detected for corticosteroids, antibiotics, drugs used in nicotine dependence, COVID-19 and 31 HPV vaccines, serotonin-norepinephrine reuptake inhibitors (SNRI), and incretin mimetics. The study purpose is clear, and the statistical methods are proper. Authors have properly addressed limitations for the study such as reporting bias. This paper will generate some values to the research community.
I don’t have major concerns for this manuscript.
The way presented in Fig 2 is uncommon. The pie chart should represent portions of a whole, meaning all parts should add up to 100 percent.
Table S1 was not attached to the manuscript.
There are a couple of typos in Table 2 and 3. The full stop or period (.) should be used as the standard decimal separator.
Author Response
Parosmia is a common problem in the era of the COVID-19 pandemic with poor results in the short-term treatment and follow-up. To investigate potential signals for drugs associated with parosmia, authors used VigiBase, the richest source of pharmacovigilance data in the world and perform disproportionality analysis. Significant disproportionate reporting was detected for corticosteroids, antibiotics, drugs used in nicotine dependence, COVID-19 and 31 HPV vaccines, serotonin-norepinephrine reuptake inhibitors (SNRI), and incretin mimetics. The study purpose is clear, and the statistical methods are proper. Authors have properly addressed limitations for the study such as reporting bias. This paper will generate some values to the research community.
We thank Reviewer 1 for his/her appreciation of our manuscript.
I don’t have major concerns for this manuscript.
The way presented in Fig 2 is uncommon. The pie chart should represent portions of a whole, meaning all parts should add up to 100 percent.
We thank Reviewer 1 for pointing this out. In fact, the Anatomical Therapeutic Chemical (ATC) classes are not mutually exclusive, so that the sum of the percentages of each class might be greater than 100 percent. We replaced the pie chart by a new graph (Figure 2), and we hope that it will meet Reviewer 1’s expectations.
Table S1 was not attached to the manuscript.
We thank Reviewer 1 for highlighting this issue of concern. We added Table S1 as an attachment to the revised manuscript and hope he/she will be able to have access to it.
There are a couple of typos in Table 2 and 3. The full stop or period (.) should be used as the standard decimal separator.
We thank Reviewer 1 for his/her careful proofreading. We updated our tables according to his/her recommendations.

Reviewer 2 Report
This manuscript investigated parosmia that appeared after administration of various drugs and Covid-19 vaccination using the WHO database. I read the manuscript with interest, especially about parosmia after the Covid-19 vaccination.
My questions are as follows;
#1. Parosmia changes the usual perception of odors, such as when the smell of something familiar is distorted or when something that usually smells pleasant now smells foul. Parosmia appears frequently following anosmia or hyposmia. Many people complain of parosmia after the recovery course of anosmia or hyposmia due to Covid-19 infection. Literature has revealed that parosmia is one of the major complaints among long-Covid symptoms.
Why did the authors exclude hyposmia in this manuscript?
Dutta S, et al. Analysis of Neurological Adverse Events Reported in VigiBase From COVID-19 Vaccines. Cureus 2022 Jan 18;14(1):e21376.
They investigated anosmia, hyposmia, and parosmia reported in VigiBase from Covid-19 vaccines.
#2. So far, hyposmia or parosmia reported after Covid-19 vaccination seems transient. Is there a possibility of long-lasting hyposmia or parosmia after the Covid-19 vaccination?
#3. Some investigators reported olfactory dysfunction after influenza vaccination.
Doty RL, et al. Influenza vaccinations and chemosensory function. Am J Rhinol Allergy 2014;28:50–53.
Nakashima T, et al. Olfactory and gustatory dysfunction caused by SARS-CoV-2: Comparison with cases of infection with influenza and other viruses. Infect Control Hosp Epidemiol 2021 Jan;42(1):113-114.
Are there any reports of olfactory dysfunction after influenza vaccination in VigiBase?
Author Response
This manuscript investigated parosmia that appeared after administration of various drugs and Covid-19 vaccination using the WHO database. I read the manuscript with interest, especially about parosmia after the Covid-19 vaccination.
We thank Reviewer 2 for his/her interest in our manuscript.
My questions are as follows;
#1. Parosmia changes the usual perception of odors, such as when the smell of something familiar is distorted or when something that usually smells pleasant now smells foul. Parosmia appears frequently following anosmia or hyposmia. Many people complain of parosmia after the recovery course of anosmia or hyposmia due to Covid-19 infection. Literature has revealed that parosmia is one of the major complaints among long-Covid symptoms.
Why did the authors exclude hyposmia in this manuscript?
We thank Reviewer 2 for helping us to clarify this point. Indeed, the main focus of our study was to assess potential iatrogenic triggers of parosmia in VigiBase®, which is a narrower scope than investigating potential drug signals associated with each olfactory dysfunction. In this context, studying and comparing these potential signals, for each olfactory dysfunction, would be the topic of a further study.
As of January 23, 2022, 1,741 cases of hyposmia (PT) were gathered in VigiBase®. The most represented ATC drug classes were Antiinfectives (525; 30,2%) and Respiratory system medications (512; 29.4%). In fact, the three leading drugs in terms of absolute number of cases were COVID-19 vaccines (384; 22.1%), oxymetazoline (197, 11.3%), and fluticasone (56; 3.2%). These drugs were subject to a disproportionality drug signal.
However, due to heterogeneity in cases coding in VigiBase®, we were not able to detect any potential preexistence of hyposmia, before the occurrence of parosmia. Furthermore, the addition of several terms in the query may lead to an increase in the risk of multiple testing bias. These results were added to the revised manuscript.
Dutta S, et al. Analysis of Neurological Adverse Events Reported in VigiBase From COVID-19 Vaccines. Cureus 2022 Jan 18;14(1):e21376.
They investigated anosmia, hyposmia, and parosmia reported in VigiBase from Covid-19 vaccines.
#2. So far, hyposmia or parosmia reported after Covid-19 vaccination seems transient. Is there a possibility of long-lasting hyposmia or parosmia after the Covid-19 vaccination?
We thank Reviewer 2 for his/her interesting question.
As of January 23, 2022, 1,741 cases of hyposmia (PT) were gathered in VigiBase®, including 384 cases (22.1%) involving COVID-19 vaccines as suspect or interacting in the occurrence of hyposmia. Among cases displaying available outcomes, 179 patients recovered or were recovering (89.4%), 89 patients did not recover (39.2%), and 5 patients (2.2%) recovered with sequelae.
Regarding parosmia and COVID-19 vaccines, most patients with available outcomes did not recover (1,124, 51.1%), while 757 (34.4%) patients recovered, 284 (25.2%) were recovering and 32 (2.8%) recovered with sequelae.
However, this analysis is hindered by missing data (e.g. the outcome is not available for each case) and a lack of follow-up of the cases. We hope that these precisions will meet the expectations of Reviewer 2. We added these findings to the manuscript.
#3. Some investigators reported olfactory dysfunction after influenza vaccination.
Doty RL, et al. Influenza vaccinations and chemosensory function. Am J Rhinol Allergy 2014;28:50–53.
Nakashima T, et al. Olfactory and gustatory dysfunction caused by SARS-CoV-2: Comparison with cases of infection with influenza and other viruses. Infect Control Hosp Epidemiol 2021 Jan;42(1):113-114.
Are there any reports of olfactory dysfunction after influenza vaccination in VigiBase?
We thank Reviewer 2 for posing this interesting issue. As of January 23, 2022, 184 cases of olfactory dysfunction following influenza vaccination were recorded in VigiBase®, among which 102 (55.4%) reports of anosmia, 75 (41.3%) reports of parosmia (Table S1) and 11 (6.0%) reports of hyposmia. However, none of these combinations with influenza vaccines were subject to any disproportionality signal. We reported the abovementioned elements regarding parosmia in the manuscript.
